# Artemisinin Attenuated Hydrogen Peroxide (H_2_O_2_)-Induced Oxidative Injury in SH-SY5Y and Hippocampal Neurons via the Activation of AMPK Pathway

**DOI:** 10.3390/ijms20112680

**Published:** 2019-05-31

**Authors:** Xia Zhao, Jiankang Fang, Shuai Li, Uma Gaur, Xingan Xing, Huan Wang, Wenhua Zheng

**Affiliations:** Center of Reproduction, Development & Aging and Institute of Translation Medicine, Faculty of Health Sciences, University of Macau, Taipa, Macau 999078, China; yb77625@um.edu.mo (X.Z.); yb57646@um.edu.mo (J.F.); yb67619@um.edu.mo (S.L.); gaur.uma2906@gmail.com (U.G.); yb77638@um.edu.mo (X.X.); yb77624@um.edu.mo (H.W.)

**Keywords:** artemisinin, SH-SY5Y cells, hippocampal neurons, H_2_O_2_, AMPK pathway

## Abstract

Oxidative stress is believed to be one of the main causes of neurodegenerative diseases such as Alzheimer’s disease (AD). The pathogenesis of AD is still not elucidated clearly but oxidative stress is one of the key hypotheses. Here, we found that artemisinin, an anti-malarial Chinese medicine, possesses neuroprotective effects. However, the antioxidative effects of artemisinin remain to be explored. In this study, we found that artemisinin rescued SH-SY5Y and hippocampal neuronal cells from hydrogen peroxide (H_2_O_2_)-induced cell death at clinically relevant doses in a concentration-dependent manner. Further studies showed that artemisinin significantly restored the nuclear morphology, improved the abnormal changes in intracellular reactive oxygen species (ROS), reduced the mitochondrial membrane potential, and caspase-3 activation, thereby attenuating apoptosis. Artemisinin also stimulated the phosphorylation of the adenosine monophosphate -activated protein kinase (AMPK) pathway in SH-SY5Y cells in a time- and concentration-dependent manner. Inhibition of the AMPK pathway attenuated the protective effect of artemisinin. These data put together suggested that artemisinin has the potential to protect neuronal cells. Similar results were obtained in primary cultured hippocampal neurons. Cumulatively, these results indicated that artemisinin protected neuronal cells from oxidative damage, at least in part through the activation of AMPK. Our findings support the role of artemisinin as a potential therapeutic agent for neurodegenerative diseases.

## 1. Introduction

Neuronal damage caused by oxidative stress (mainly reactive oxygen species) is one of the major causes of neurodegenerative diseases such as Alzheimer’s disease (AD) [1,2,3]. Excessive levels of reactive oxygen species (ROS) enhances cellular oxidative stress, leading to lipid peroxidation, protein denaturation, and DNA damage, disrupting cell function and integrity, leading to cell apoptosis [4]. Oxidative stress is an imbalance between ROS production and antioxidant defense and has been found to be associated with aging and aging-related neurodegenerative diseases [5,6,7]. H_2_O_2_ is an important source of ROS and is widely used as an inducer of oxidative stress in cell models for the study of various neurodegenerative diseases caused by oxidative stress [8,9].

Artemisinin is extracted from the plant *Artemisia annua*. It is one of the most effective anti-malarial drug which has saved the lives of millions of malaria patients worldwide [10,11]. In addition to anti-malarial benefits, artemisinin has many other biological and pharmacological properties, including antioxidant, anti-inflammatory, antiviral and anti-tumor effects [12,13,14]. In addition, we have recently shown that artemisinin have neuroprotective potential [15,16,17]. However, the effect and underlying action mechanism of artemisinin against oxidative stress in SH-SY5Y neuronal cells and primary cultured neurons are still unknown. Artemisinin can cross the blood–brain barrier (BBB) and has no observable toxicity to the central nervous system itself. This support the hypothesis that artemisinin can have a favorable effect in the treatment of neurological diseases [18]. 

Decreased mitochondrial membrane potential is a marker of early cell apoptosis and is closely related to cellular oxidative damage and apoptosis.The production of ROS occurs mainly in mitochondria and it accumulates during aging ultimately becoming a major cause of cellular damage. AMP-activated protein kinase (AMPK) is an energy sensor which plays a key role in regulating complex signaling networks of mitochondrial biogenesis [19]. Emerging evidence have shown that AMPK not only maintains energy metabolism balance, but also regulates oxidative stress, endoplasmic reticulum stress, autophagy and inflammation, thereby increasing stress resistance [20]. Moreover, AMPK activation seems to decline during aging. In addition, activation of the AMPK cascade have been shown to be associated with improved glucose and lipid metabolism, as well as neuroprotective and anticancer effects [21,22]. The effects of age associated decline in AMPK activity on mitochondrial dysfunction and age related oxidative damage have also been verified previously [23].

In the present study, we evaluated whether artemisinin can protect against H_2_O_2_-induced cytotoxicity and the potential mechanisms behind the protective effects of artemisinin in SH-SY5Y cells and primary hippocampal neurons. Our results showed that artemisinin protected SH-SY5Y cells from H_2_O_2_-induced injury as indicated by3-(4,5-dimethylthiazol-2-yl)-2,5-diphenyl tetrazolium bromide (MTT) assay and mitochondrial membrane potential assay. Furthermore the nuclear morphology and abnormal changes in intracellular ROS were also restored to normal by artemisinin treatment. In addition, caspase-3 which is a key performer of apoptosis and was reported to be activated by H_2_O_2_ [24], was shown to be downregulated upon artemisinin treatment in present study. The AMPK signaling pathway plays an important role in systemic energy balance and metabolism and regulation of age-related diseases [11,25,26]. Finally, we also delineated the role of the AMPK signaling pathway in the protective effect of artemisinin. All of these may provide interesting insights into the potential applications of artemisinin in future AD research.

## 2. Results

### 2.1. Artemisinin Attenuated the Cell Viability Induced by H_2_O_2_ in SH-SY5Y Cells

SH-SY5Y cells were incubated with different concentrations of artemisinin or H_2_O_2_ for 24 h and the cell viability was determined by MTT assay. No cytotoxic effect was observed upon artemisinin treatment in SH-SY5Y cells up to 200 μM concentration (Figure 1B). Compared with the control group, H_2_O_2_ caused significant cytotoxicity in SH-SY5Y cells 600 μM onwards (Figure 1C), and therefore 600 μM H_2_O_2_ concentration was used in further experiments. To test the protective effects of artemisinin, SH-SY5Y cells were pretreated with artemisinin for 2 h before being exposed to H_2_O_2_ for another 24 h. The result showed that pre-treatment with artemisinin significantly reduced H_2_O_2_-induced cell death (Figure 1D) and 12.5 μM artemisinin was used in further experiments. 

### 2.2. Artemisinin Pretreatment Attenuated H_2_O_2_-Induced Apoptosis in SH-SY5Y Cells

Both apoptosis and necrosis contribute to the cell viability loss during cell injuries. We tested if the protective effect of artemisinin against H_2_O_2_ insult was mediated by its anti-apoptosis effects. Nuclei condensation was observed in SH-SY5Y cells after exposure to 600 μM H_2_O_2_ in Hoechst 33342 staining assay. However, pre-treatment with 12.5 μM artemisinin significantly improved these changes (Figure 2A,B). The result was further confirmed using flow cytometry for Annexin V-FITC/PI-positive cells and data from these experiments indicated that H_2_O_2_ exposure markedly increased apoptosis in SH-SY5Y cells, while 12.5 μM artemisinin pretreatment significantly reduced the apoptosis caused by H_2_O_2_ (Figure 2C,D). Caspase-3 plays an important role in apoptosis and in order to further verify the anti-apoptosis effect of artemisinin we checked the caspase-3 activity. We found that artemisinin reversed H_2_O_2_-induced increase in the activity of caspase—(Figure 2E). 

### 2.3. Artemisinin Inhibited H_2_O_2_-Induced Increase in ROS Level and Restored the Mitochondrial Membrane Potential in SH-SY5Y Cells

Loss of mitochondrial membrane potential (∆ψm) due to mitochondrial inhibition was involved in the cell apoptosis caused by H_2_O_2_. In further study, we elucidated whether artemisinin could reduce H_2_O_2_-induced ∆ψm loss. The ∆ψm in SH-SY5Y cells was assessed by analyzing the red/green fluorescent intensity ratio of JC-1 staining. The results revealed that artemisinin pretreatment significantly prevented the decline of ∆ψm induced by H_2_O_2_ (Figure 3A,C). The generation of excess ROS is considered to be among the main causes of cell apoptosis induced by H_2_O_2_. Therefore, we investigated whether artemisinin blocked H_2_O_2_-induced oxidative stress in SH-SY5Y cells. Cellular oxidative stress was determined by the CellROXs Deep Red Reagent. SH-SY5Y cells pretreated with or without 12.5 μM artemisinin for 2 h were treated with 600 μM H_2_O_2_ for 24 h. As expected (Figure 3B,D), artemisinin significantly decreased the intracellular ROS production induced by H_2_O_2_. 

### 2.4. Artemisinin Stimulated the Phosphorylation of AMPK in SH-SY5Y Cells

AMPK is a highly conserved regulator of cellular energy metabolism that plays an important role in regulating cell growth, proliferation, survival, and regulation of energy metabolism in the body. We therefore tested whether AMPK is involved in protective effect of artemisinin in SH-SY5Y cells. As shown in Figure 4, after treatment with different doses of artemisinin for different time points, the phosphorylation of AMPK was gradually increased (Figure 4A–D).

### 2.5. AMPK Mediated the Protective Effects of Artemisinin in SH-SY5Y Cells

In order to examine whether AMPK is involved in the survival promoting effect of artemisinin on cell apoptosis induced by H_2_O_2_, we knocked down AMPKα by using shRNA plasmid specific for AMPK in SH-SY5Y cells (Figure 5A). After 24 h of transfection, cells were treated with or without artemisinin, then incubated with 600 μM H_2_O_2_ for 24 h, and the cell viability was determined by MTT assay. As shown in Figure 5B, knockdown of AMPKα by shRNA significantly attenuated the protective effect of artemisinin. To further confirm the effect of AMPK, we used the compound C (a specific inhibitor for the AMPK) to pretreat cells for 30 min. MTT assay showed that pretreatment with Compound C blocked the protective effects of artemisinin (Figure 5C). We got the similar result from TUNEL staining assay (Figure 5D,E). Western blot also showed that AMPK pathway played an important role in the protective effect of artemisinin (Figure 5G,H). At the same time, blocking the AMPK signaling pathway also restored the activity of caspase-3 (Figure 5G,I).

### 2.6. Artemisinin Protected Primary Cultured Hippocampal Neurons Against H_2_O_2_ Induced Injury via AMPK Kinase

In order to verify whether the neuroprotective effects of artemisinin against H_2_O_2_ are limited to SH-SY5Y, we also checked its protective effects on primary cultured hippocampal neurons. In concordance with SH-SY5Y, artemisinin successfully protected hippocampal neurons against H_2_O_2_ insult in a concentration-dependent manner as shown in Figure 6A–C and Appendix A. Similar results was found in cortex neuron (Appendix A).The primary cultured neurons were more sensitive to both H_2_O_2_ insult and artemisinin treatment and therefore artemisinin achieved its significant protective effect at a lower concentration. Moreover, the protective effect of artemisinin is also inhibited by AMPK inhibitor compound C in these primary neuron cells (Figure 6E–H). The protein NeuN is localized in the nucleus and perinuclear cytoplasm of most neurons in the central nervous system. It is widely used to assess the functional status of neurons and their pathological states. Our results show that artemisinin can improve neuronal damage caused by H_2_O_2_ (Figure 6D). These results are consistent with our results from SH-SY5Y cells and further confirmed that artemisinin was able to protect neuronal cells from oxidative stress via the activation of AMPK kinase.

## 3. Discussion

In this study, we demonstrated that artemisinin blocked H_2_O_2_-induced neuronal damage by activating the AMPK pathway. This study discovered a novel neuroprotective effect of artemisinin, suggesting that artemisinin may have potential for the treatment of neurodegenerative diseases. In addition to antimalarial effect, recently, artemisinin has also demonstrated to possess anti-tumor and anti-inflammatory properties [27]. The hippocampus is one of the most vulnerable parts of the brain and is susceptible to numerous pathological conditions [28]. In the current study, we used H_2_O_2_ injury cellular model SH-SY5Y and hippocampal neurons to study the effects of artemisinin on oxidative stress. We found that pretreatment with artemisinin provided protection to SH-SY5Y and hippocampal neurons from H_2_O_2_-induced damage. Consistent with others and our previous reports, we found that H_2_O_2_ is cytotoxic to SH-SY5Y cells in a dose-dependent pattern [29]. In addition, data from Hoechst 33342 staining assay and flow cytometry showed that H_2_O_2_ also induced apoptosis in SH-SY5Y cells. The cell viability of SH-SY5Y cells incubated with H_2_O_2_ was significantly increased when pretreated with artemisinin. In addition, artemisinin can also attenuate H_2_O_2_-induced apoptosis of SH-SY5Y cells, which further suggests that artemisinin has a protective effect on H_2_O_2_-induced SH-SY5Y cells.

Mitochondria are the major site of ROS production [30], and H_2_O_2_ increases oxidative stress damage by increasing ROS production [29]. We concluded that pretreatment with artemisinin reduced H_2_O_2_-induced ROS accumulation. Mitochondrial dysfunction caused by oxidative stress plays a key role in the pathogenesis of aging-related neurodegenerative diseases [31]. Strategies to block mitochondrial dysfunction have been declared to be a potential therapy for preventing cell death [32]. The results of the current study have shown that artemisinin pretreatment can inhibit H_2_O_2_-induced ∆ψm loss.

The AMPK signaling pathway plays a major role in cell survival, apoptosis and senescence prevention [33]. This pathway has been shown to be a key signaling pathway that induces cellular antioxidant mechanisms. Our data suggest that artemisinin pretreatment stimulates phosphorylation of AMPK and activates the AMPK pathway and plays a key role in the neuroprotective effects of H_2_O_2_ induced injury, consistent with activation of the AMPK cascade. The protective effect of artemisinin was attenuated after blocking the AMPK signaling pathway with the specific knockout plasmids of AMPK and specific pharmacological inhibitor Compound C. Consistently, pretreatment with artemisinin reversed [34] the increase in caspase-3 activity caused by H_2_O_2_. After blocking AMPK, the effect of artemisinin on caspase-3 activity disappeared. A major mechanism in the cellular defense against oxidative is activation of the Nrf2-antioxidant response element signaling pathway, but when blocked AMPK pathway Nrf2 expression was decreased (Appendix A).Taken together, these results provided mechanistic evidence to support the view that artemisinin-mediated protection against H_2_O_2_ induced injury occurs through AMPK activation. We got consistent results in hippocampal neurons as well (Figure 6). Our results suggest that the possible regulatory mechanism of artemisinin operates by protecting neuronal SH-SY5Y cells and hippocampal neurons from H_2_O_2_-induced oxidative damage via activating the AMPK signaling pathway (Figure 7), including inhibition of intracellular ROS production; restoration of mitochondrial membrane potential (∆ψm); activation of AMPK signaling pathway; and reduction of caspase-3 activity. In addition, we also checked some of the other derivatives of artemisinin, like dihydroartemisinin (DHA), performed cellular viability assays, and found that DHA was less effective. The results are attached in the Appendix A. When we compared both the compound cellular viability assay results we found that the artemisinin was better than DHA. Owing to its lipid-soluble nature, artemisinin can pass the blood brain barrier and maintain a higher concentration in the central nervous system [35], giving it advantage over other neurological drugs. Also as an FDA-approved anti-malarial drug, artemisinin has been clinically used for decades with no significant adverse effects [36].

Our results found that artemisinin can eliminate ROS and protect the SH-SY5Y and hippocampal neurons from H_2_O_2_-induced oxidative damage. Therefore, artemisinin has the potential to be a novel drug to prevent and treat neurodegenerative diseases.

## 4. Materials and Methods

### 4.1. Materials

Analytical grade artemisinin was purchased from Meilunbio (Dalian China). Dimethyl sulfoxide (DMSO), Dulbecco’s modified Eagle’s medium (DMEM) and hydrogen peroxide (H_2_O_2_) were procured from Sigma (St. Louis, MO, USA). Poly-D-lysine, 3-(4,5-Dimethylthiazol-2-yl)-2,5-diphenyltetrazolium bromide (MTT), 5,5′,6,6′-tetrachloro-1,1′,3,3′-tetraethyl-benzimidazolyl-carbocyanineiodide (JC-1) and Hoechst 33258 were purchased from Molecular Probes (Eugene, OR, USA). anti-phospho-AMPK (2525, 1:1000 Rabbit IgG), anti-AMPK (5831, 1:1000 Rabbit IgG), NeuN (24307, 1:100 Rabbit IgG) and GAPDH antibodies (2683s, 1:1000 Rabbit IgG) were purchased from Cell Signaling Technology (Woburn, MA, USA). CellROX Deep Red Reagent were ordered from Thermo Fisher Scientific (Rockford, IL, USA). Annexin V-FITC/PI Apoptosis Detection Kit was obtained from BD Biosciences (San Diego, CA, USA). Fetal bovine serum (FBS) and 0.25% Trypsin were purchased from Life Technologies (Grand Island, NY, USA). AMPK-ShRNA was purchased from Shanghai Genechem Co; Ltd. (Shanghai, China). AMPK inhibitor (Compound C) was ordered from Calbiochem (San Diego, CA, USA).

### 4.2. Methods

#### 4.2.1. SH-SY5Y Cell Culture

Human neuroblastoma SH-SY5Y cells were cultured in 75-cm^2^ flasks in DMEM supplemented with 10% heat-inactivated FBS and 0.1% penicillin/streptomycin at 37 °C with 5% CO_2_ humidified atmosphere. The medium was replaced every 2–3 days, and cells were sub-cultured once 80–90% confluency was reached. After digestion with 0.25% trypsin, cells were collected by centrifugation at 1000 rpm for 5 min and resuspended in fresh medium. Cells were seeded into 96-well, 12-well or six-well plates and grown overnight. Adherent cells were used for further experiments. 

#### 4.2.2. Hippocampal Neurons Culture

Newborn C57BL/6 mice were procured from the animal facility of University of Macau. The whole body was disinfected with 75% alcohol and the brain was surgically removed and stored into cold HBSS (Ca^2+^, Mg^2+^ free) balance solution. The whole hippocampus region was dissected using a glass rod which was bent on both sides. The hippocampus was cleared of the blood and the mixed blood vessels by washing thrice with HBSS. Then the hippocampus was chopped into 1 mm^3^ pieces using scissors and after washing thrice with HBSS the tissue was digested with 0.125% of trypsin at 37 °C for 15 min. The enzymatic digestion was stopped with 10% FBS and 5 mL of Neurobasal A (Gibco, USA) was added to the digested hippocampus tissue in a 15 mL centrifuge tube. The turbid tissue supernatant was collected in another 15 mL centrifuge tube and centrifuged at 1000 rpm for 10 min. The resulting cell pellet was resuspended in Neurobasal A/B27 (Gibco, Carlsbad, CA, USA) and seeded in poly-D-lysine treated plates at a density of about 1–2 × 10^5^ cells / mL and incubated for growth at 37 °C in 5% CO_2_ humidified atmosphere.

#### 4.2.3. MTT Assay

For the MTT assay, SH-SY5Y cells were seeded at a density of 6–8 × 10^3^ cells/well in 96-well plates with 1% FBS medium for 24 h. After serum starvation, the cells were incubated with drugs or inhibitors for appropriate time, and treated with H_2_O_2_ for another 24 h. The cells were then incubated with MTT (0.5 mg/mL) for additional 3–4 h. The medium was aspirated from each well and DMSO (100 μL) was added to dissolve the dark-blue formazan crystals that were formed in intact cells and absorbance of each well solution was measured with a microplate reader (SpectraMax 250, Molecular Device, Sunnyvale, CA, USA). The data were presented as Optical Density (OD) at a wavelength of 570 nm. 

#### 4.2.4. Hoechst 33258 Staining

Apoptosis of cells was examined by staining with the DNA binding dye. SH-SY5Y cells were pretreated with 12.5 μM artemisinin for 2h before being exposed to 600 μM H_2_O_2_ for another 24 h, followed by fixing the cells in 4% formaldehyde in PBS for 10 min at 4 °C. The fixed cells were further incubated for 10 min with 10 µg/mL of Hoechst 33258 in order to stain the nuclei. After rinsing twice with PBS, the apoptotic cells were visualized under a fluorescent microscope (Olympus, Japan). Cells exhibiting apoptosis hallmarks such as condensed chromatin or fragmented nuclei were scored as apoptotic cells. A minimum of 200 cells in five random fields were collected and quantified for each Hoechst staining experiment. The data was statistically presented as percentage of apoptotic cells.

#### 4.2.5. Annexin V-FITC/PI Staining for Apoptosis Evaluation

Flow cytometry using Annexin V-FITC/PI staining was carried out for apoptosis evaluation. Flow cytometry was performed as described in the guidelines of assay kit. Briefly, SH-SY5Y cells were pretreated with 12.5 μM artemisinin for 2 h before being exposed to 600 μM H_2_O_2_ for another 24 h, then the cells were trypsinized and washed twice with ice-cold PBS then centrifuged at 1000 rpm for 5 min and re-suspended in Annexin V-FITC/PI binding buffer (195 μL). Annexin V-FITC (5 μL) was supplemented and the cells were incubated in the dark at room temperature for 10 min. The cells were then centrifuged at 1000 rpm for 5 min and re-suspended in Annexin V-FITC/PI binding buffer (190 μL); propidium iodide (PI) (10 μL) was further added, followed by incubation in the dark for 5 min. Apoptosis was quantified using Flow cytometry. Cell Quest Pro software (BD AccuriC6, BD, USA) was used for analyzing apoptosis condition.

#### 4.2.6. Measurement of Intracellular ROS Levels

Intracellular ROS generation was assessed by CellROXs Deep Red Reagent. After pretreatment with 12.5 μM artemisinin for 2 h before being exposed to 600μM H_2_O_2_ for another 24 h, the SH-SY5Y cells were exposed to CellROXs Deep Red Reagent (5 mM) in DMEM for 1 h in the dark. After rinsing twice with 1× PBS solution, the Fluorescence was observed and recorded on a fluorescent microscope at excitation and emission wavelengths of 640 nm and 665 nm respectively. The semi quantification of ROS level was analyzed using Image-J software (version 1.48; NIH, Bethesda, MD, USA) and all the values of ROS levels were normalized to the control group.

#### 4.2.7. Measurement of Mitochondrial Membrane Potential (∆ψm)

JC-1 dye was utilized to monitor mitochondrial integrity. In brief, SH-SY5Y cells were seeded into 96-well plates (1 × 10^4^ cells/well) in dark. After pretreatment with 12.5 μM artemisinin for 2 h before being exposed to 600 μM H_2_O_2_ for another 24 h, the cells were treated with JC-1 dye (10 μg/mL in medium) for 15 min at 37 °C and rinsed twice with PBS. For quantification of the signal, the intensities of red (excitation 560 nm, emission 595 nm) and green fluorescence (excitation 485 nm, emission 535 nm) were assessed using an Infinite M200 PRO Multimode Microplate reader. ∆ψm was calculated as the ratio of red/green fluorescence intensity and the values were normalized with respect to the control group. The fluorescent signal in the cells was also recorded with a fluorescent microscope.

#### 4.2.8. Immunocytochemistry (ICC)

ICC is a method of detecting specific antigens in cells using an appropriate antibody labeling strategy. After drug treatment, cells were washed twice in PBS and then fixed in 4% paraformaldehyde for 15 min at room temperature to maintain cell morphology. The cells were then rinsed three times with PBS, and incubated in PBST (0.1% Triton X-100 in 1× PBS) for 20 min at room temperature. Following this, the cells were blocked with 1% BSA for 1 h at room temperature which can help reduce non-specific hydrophobic interactions. And then the primary antibody/antibodies were added and kept at 4 °C overnight. The next day after washing cells with PBS three times, the fluorophore-conjugated secondary antibodies were added for 2 h at room temperature, away from light. A drop of mounting medium was added to each slide. Then, samples were observed by confocal laser scanning microscopy.

#### 4.2.9. TUNEL Assay

SH-SY5Y cells were pre-treated with 5 μM Compound C (AMPK inhibitor) for 30 min, and treated with 12.5 μM artemisinin for 2 h, then incubated with or without 600 μM H_2_O_2_ for a further 24 h. After treatment, the cells were fixed with 4% PFA for 30 min and washed with 0.1% Triton-X PBS for 3 times. Then cells were incubated with TUNEL test solution (C1086, Beyotime, China) at 37 °C for 60 min in the dark following the manufacturer’s instruction. After washing with 1× PBS, images were taken with a fluorescence microscope.

#### 4.2.10. Caspase-3 Activity Assay

After drug treatment, SH-SY5Y cells were digested with trypsin and harvested by centrifugation at 600 *g* for 5 min at 4 °C. The supernatant was carefully aspirated and washed once with PBS. According to the manufacturer’s protocol, 100 μL of lysate was added per two million cells, and the pellet was resuspended. The supernatant was transferred to an ice-cold centrifuge tube and lysed on ice for 15 min. Then centrifuged at 16,000–20,000× *g* for 15 min at 4 °C. The reaction system was set up according to the manufacturer’s instructions. The detection buffer was added first followed by the sample to be tested, and finally 10 μL of Ac-DEVD-pNA (2 mM) was added, mixed well avoiding the bubbles. The reaction system was then incubated in a working solution for 60–120 min at 37 °C. A405 was then determined by Infinite M200 PRO Multimode Microplate. All values of caspase-3 activities were normalized to the control group.

#### 4.2.11. Western Blotting

Western blotting was performed following the standard procedure [15]. The experimental cells under different treatments were lysed in ice-cold RIPA lysis buffer and the protein concentration was assessed using a BCA protein assay kit according to the manufacturer’s instructions. Samples with the same amount of proteins were separated on 10% polyacrylamide gels, and later transferred to PVDF membrane. After blocking with 3% BSA for 1 h, the membranes were incubated with selective primary antibodies overnight. Following day, the primary antibody was washed with 1× TBST thrice, and incubated with secondary antibody for another 2 h. After exposure with BCL, the intensity of the bands was quantified using Image J software.

#### 4.2.12. Transfection of ShRNA Plasmid

The shRNA of AMPKα were designed and synthesized by GenPharma Co., Ltd. (Shanghai, China). One day before the transfection, SH-SY5Y cells were seeded into a 12-well plate at a density of 1 × 10^5^ cells/well, grown overnight, and transfected when the cell density reached 80%. SH-SY5Y cells were transfected with Lipofectamine 2000 reagent according to the manufacturer’s protocol. Protein samples were collected after 48 h of transfection. The AMPKα knockdown efficiency was verified by western blot.

#### 4.2.13. Data Analysis and Statistics

Statistical analysis was performed using GraphPad Prism 5.0 statistical software (GraphPad software, Inc., San Diego, CA, USA). All experiments were performed in triplicates. Data are expressed as mean ± standard deviation (SD). Statistical analysis was carried out using one-way ANOVA followed by Tukey’s multiple comparison, with *p* < 0.05 considered statistically significant.

## Figures and Tables

**Figure 1 ijms-20-02680-f001:**
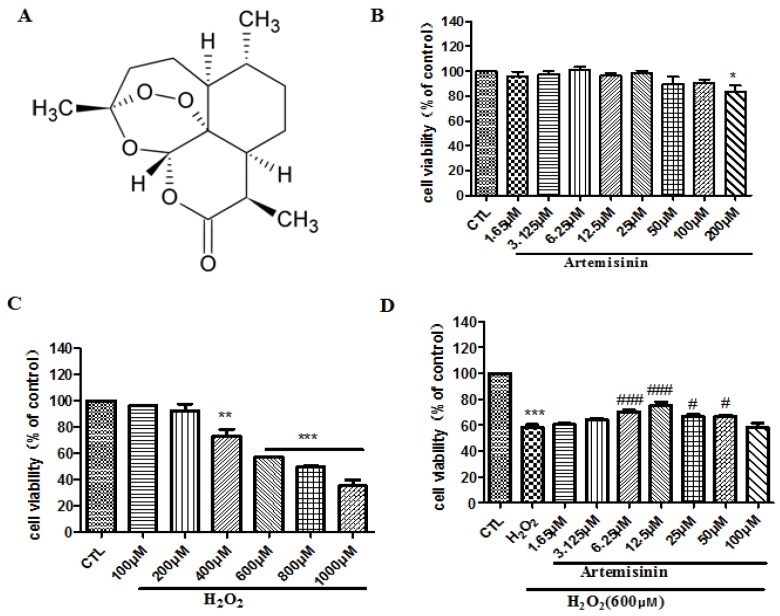
Artemisinin attenuated the decrease in cell viability caused by H_2_O_2_ in SH-SY5Y cells. (**A**) The structure of artemisinin. (**B**) The cytotoxicity of artemisinin, cells were treated with artemisinin (1.65–200 μM) for 24 h and cell viability was measured using the MTT(3-(4,5-dimethylthiazol-2-yl)-2,5-diphenyl tetrazolium bromide) assay. (**C**) The cytotoxicity of H_2_O_2_. (**D**) Cells were pretreated with artemisinin at indicated concentrations and then induced with or without 600 μM H_2_O_2_ for a further 24 h and cell viability was measured using the MTT assay. Data represent means ± SD, * *p* < 0.05, ** *p* < 0.01, *** *p* < 0.001, ^#^
*p* < 0.05, ^###^
*p* < 0.001.

**Figure 2 ijms-20-02680-f002:**
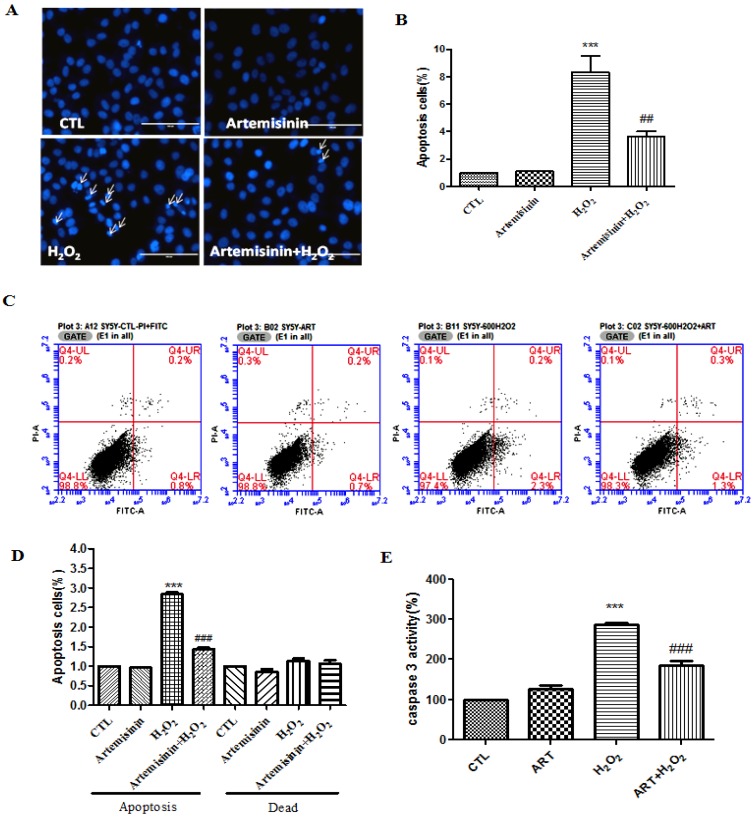
Artemisinin suppressed H_2_O_2_-induced apoptosis in SH-SY5Y cells. Cells were pre-treated with 12.5 μM artemisinin for 2 h and then induced with or without 600 μM H_2_O_2_ for another 24 h. The pictures have been taken at a magnification of 40× (100 μm). (**A**) Photographs of representative cultures measured by Hoechst staining. Apoptotic cells are marked with white arrows (**B**) Quantitative analysis of (**A**). (**C**) Photographs of representative cultures measured by flow cytometry. (**D**) Quantitative analysis of (**C**). (**E**) The activity of caspase-3 was monitored by caspase assay. Data represent means ± SD, *** *p* < 0.001, ^##^
*p* < 0.01, ^###^
*p* < 0.001.

**Figure 3 ijms-20-02680-f003:**
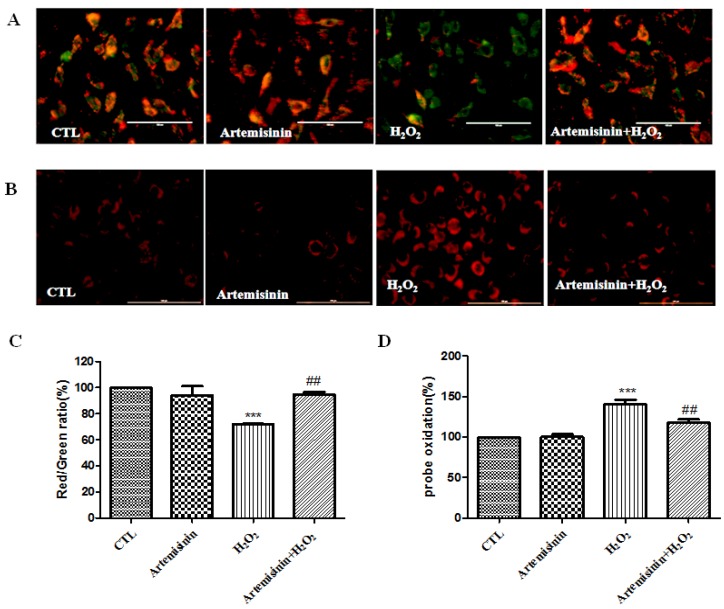
Artemisinin inhibited H_2_O_2_-induced increase of reactive oxygen species (ROS) level and restored the mitochondrial membrane potential in SH-SY5Y cells. (**A**). After pre-treatment with 12.5 μM artemisinin or 0.1% DMSO (vehicle control) for 2 h, SH-SY5Y cells were incubated with or without 600 μM H_2_O_2_ for another 24 h. The decline in the membrane potential was reflected by the shift of fluorescence from red to green indicated by JC-1. The pictures have been taken at a magnification of 40× (100 μm). (**B**). Intracellular ROS level was measured by the CellROXs Deep Red Reagent. The pictures has been taken on 40× (100 μm). (**C**). Quantitative analysis of (**A**). (**D**). Quantitative analysis of (**B**). The data were represented as the mean ± SD of three independent experiments. *** *p* < 0.001, ^##^
*p* < 0.01.

**Figure 4 ijms-20-02680-f004:**
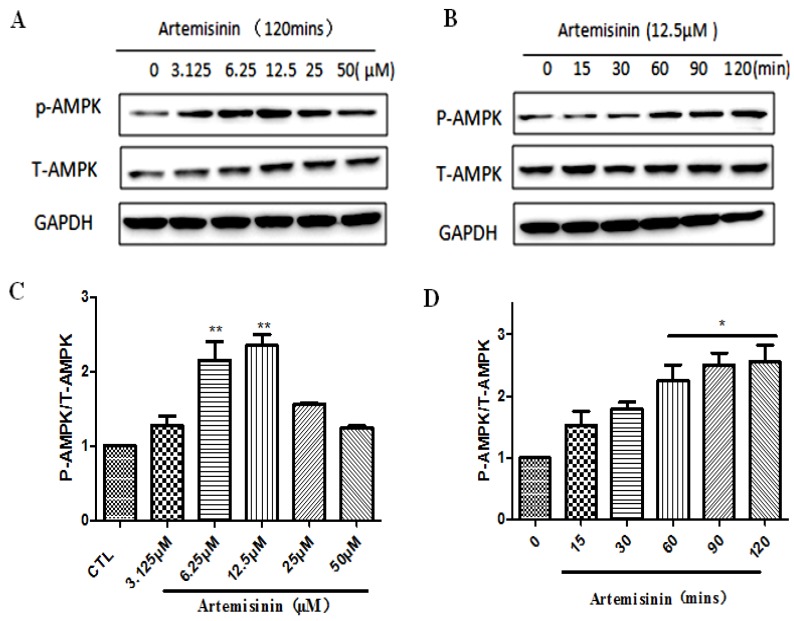
Artemisinin stimulated the phosphorylation of AMP-activated protein kinase (AMPK) in SH-SY5Y cells. (**A**,**C**) The SH-SY5Y cells were collected with artemisinin treatment for different times (0, 15, 30, 60, 90 and 120 min) at 12.5 μM, and at different concentrations (3.15, 6.25, 12.5, 25 and 50 μM) for 120 min. The expression of phosphocreatine AMPK, total AMPK and glyceraldehyde-3-phosphate dehydrogenase (GAPDH) were detected by western blot. (**B**,**D**) Quantification of representative protein band from western blotting. The data were represented as the mean ± SD. * *p* < 0.05, ** *p* < 0.01, versus the control group was considered significantly different.

**Figure 5 ijms-20-02680-f005:**
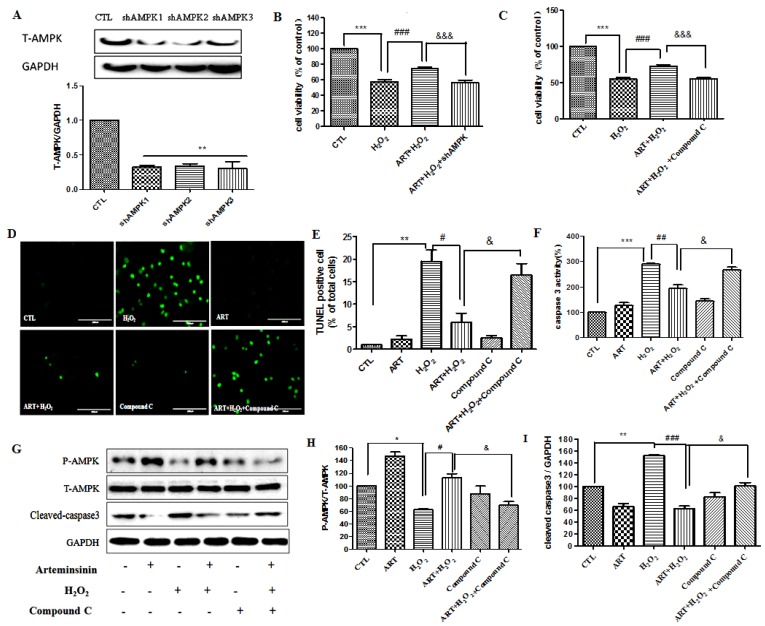
AMPK pathway mediated the protective effects of artemisinin in SH-SY5Y cells. (**A**). Cells were pretreated with shAMPKα 1–3 plasmid for 24 h, The knockout efficiency was determined by western blot. (**B**). Cells were pretreated with shAMPKα1 plasmid for 24 h before 12.5 μM artemisinin treatment for 2 h, and then incubated with or without 600 μM H_2_O_2_ for a further 24 h. Cell viability was evaluated using the MTT assay. (**C**). Cells were pre-treated with 5 μM Compound C (AMPK inhibitor) for 30 min, and treated with 12.5 μM artemisinin for 2 h, then incubated with or without 600 μM H_2_O_2_ for another 24 h, and the cell viability was determined by MTT assay. (**D**,**E**). TUNEL staining manifested that artemisinin attenuated H_2_O_2_-induced cell apoptosis significantly. The pictures have been taken at a magnification of 40× (100 μm). (**F**). The activity of caspase-3 was measured by caspase assay. (**G**). The expression of phosphorylated AMPK, total AMPK, cleaved caspase-3 and GAPDH were detected by western blot. (**H**,**I**). Quantitative analysis of (**G**). The data was represented as the mean ± SD. * *p* < 0.05, ** *p* < 0.01, *** *p* < 0.001, ^##^
*p* < 0.01, ^###^
*p* < 0.001, ^&^
*p* < 0.05, ^&&&^
*p* < 0.001.

**Figure 6 ijms-20-02680-f006:**
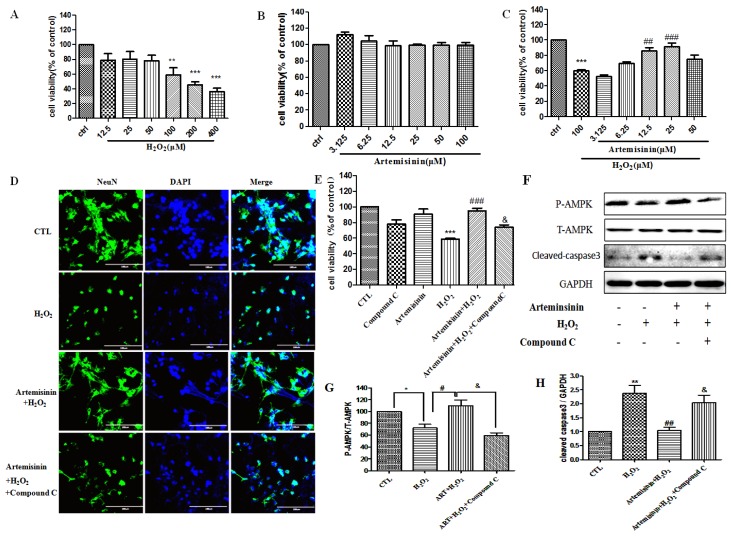
Artemisinin attenuated the decrease in cell viability caused by H_2_O_2_ in neuronal cells. (**A**) The cytotoxicity of H_2_O_2_ on neuronal cells. (**B**) Dose of artemisinin. (**C**) Primary cultured hippocampal neurons were pretreated with artemisinin at indicated concentrations and then induced with or without 100 μM H_2_O_2_ for another 24 h, and cell viability was measured using the MTT assay. (**D**) Primary cultured hippocampal neurons pretreated with 2.5 μM compound C for 30 min were treated with 25 μM artemisinin for 2 h, and then incubated with or without 100 μM H_2_O_2_ for another 24 h. Immunocytochemistry of NeuN in each group was detected. The pictures were taken at a magnification of 40× (100 μm). (**E**) Primary cultured hippocampal neurons pretreated with 2.5 μM compound C for 30 min were treated with 25 μM artemisinin for 2 h, and then incubated with or without 100 μM H_2_O_2_ for another 24 h. Cell viability was measured by the MTT assay. (**F**) The expression of phosphorylated AMPK, total AMPK, cleaved caspase-3 and GAPDH were detected by western blot. (**G**,**H**) Quantitative analysis of (**F**). The data were represented as the mean ± SD. ** *p* < 0.01, *** *p* < 0.001, ^##^
*p* < 0.01, ^###^
*p* < 0.001.

**Figure 7 ijms-20-02680-f007:**
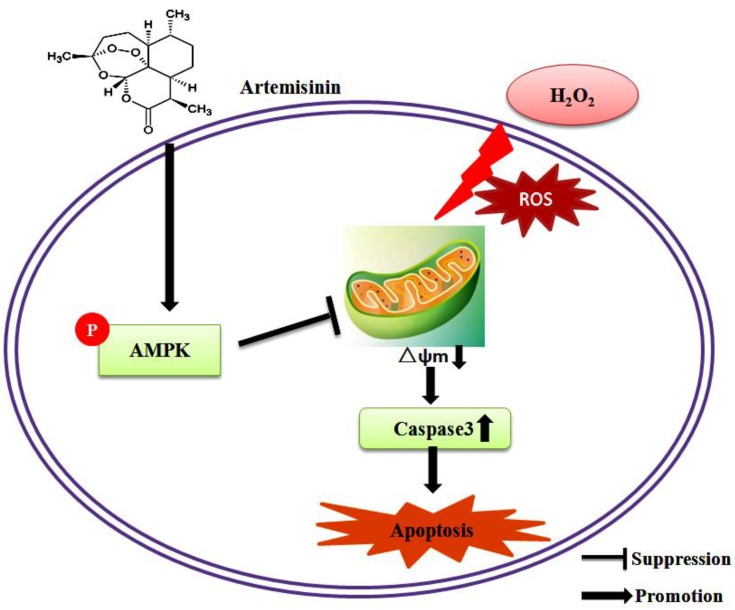
The possible mechanism of artemisinin. Excessive H_2_O_2_ leads to excessive ROS production and abnormal mitochondrial membrane potential, which in turn produces oxidative stress. Oxidative stress further leads to activation of caspase-3, which ultimately initiates apoptosis. Administration of artemisinin significantly inhibited H_2_O_2_-induced apoptosis and artemisinin ameliorated abnormal changes in these markers by activating the AMPK pathway. By blocking AMPK, the protective effect of artemisinin disappears. Therefore, artemisinin can be considered as a promising candidate for the treatment of neurodegenerative diseases caused by oxidative stress.

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
