# Peer review of "Artemisinin Attenuated Hydrogen Peroxide (H2O2)-Induced Oxidative Injury in SH-SY5Y and Hippocampal Neurons via the Activation of AMPK Pathway"

_ijms, 2019, doi:10.3390/ijms20112680_

Round 1

Reviewer 1 Report

Review of manuscript ijms-505223

In this paper the authors study the potential neuroprotective effects of the natural product artemisinin in cell culture models. They use a neuroblastoma cell line and primary hippocampal cells to test the protective effects of this substance. They use a variety of techniques and it is a very through set of studies aimed at showing protective effects against oxidative stress and show some evidence that this is mediated through the AMPK pathway. Overall it is a lot of data, and contributes to the scientific literature in this field. I do have a few suggestions that I believe should be addresses before publication:

1.There are many grammatical mistakes and therefore careful editing by a native English speaker is needed. For example, in the abstract on line 15, hypothesis should be ‘hypotheses’ or it should be written  ‘…oxidative stress is a key hypothesis’. Also in the abstract ‘the’ is used many times where inappropriate. These are just a few of many examples throughout the manuscript.

2.On the first line of the intro the authors state that oxidative stress is one of the major causes of neurodegenerative diseases. This is too strong of a statement, and it should state that oxidative stress ‘is believed to be’ one of the major causes of neurodegenerative diseases.

3.In figure 1C some of the bars don’t seem to have standard deviation bars (100uM and 600uM). Is the data really represented accurately or was there a very small n?

4.For the data in Figure 2A-B, were the cells counted by more than one observer, or just one person? Counting apoptotic cells can be somewhat subjective so the average counts from at least two individuals should be done, and if this was the case it should be mentioned in the methods.

5.Ideally it would have been nice to see more experiments in addition to cell viability in hippocampal cells. The sensitivity to H2O2 and artemisinin was increased in these cells compared to SH-SY5Y cells, so do the authors expect that other results would be changed (e.g. AMPK pathway activation)?

Author Response

Response to reviewer 1 comments

Comments/Suggestions:

1. There are many grammatical mistakes and therefore careful editing by a native English speaker is needed. For example, in the abstract on line 15, hypothesis should be ‘hypotheses’ or it should be written ‘…oxidative stress is a key hypothesis’. Also in the abstract ‘the’ is used many times where inappropriate. These are just a few of many examples throughout the manuscript.

Response: Thanks a lot for your comments. The manuscript has been carefully edited by a native English speaker according to the suggestion.

2. On the first line of the intro the authors state that oxidative stress is one of the major causes of neurodegenerative diseases. This is too strong of a statement, and it should state that oxidative stress ‘is believed to be’ one of the major causes of neurodegenerative diseases.

Response: Thanks a lot for reviewer suggestion. We have revised as suggested.

3. In figure 1C some of the bars don’t seem to have standard deviation bars (100uM and 600uM). Is the data really represented accurately or was there a very small n?

Response: Thanks the reviewer for the comments. We performed this experiment using a 96-well plate and repeated the experiment three times with 6 replicates each time. As we count cell carefully each time and used the average of three experiments so the difference within the group is not large, and some groups have small bar value.

4. For the data in Figure 2A-B, were the cells counted by more than one observer, or just one person? Counting apoptotic cells can be somewhat subjective so the average counts from at least two individuals should be done, and if this was the case it should be mentioned in the methods.

Response: Thank the reviewer for the comments. The reviewer's advice is good and more reasonable. We have done three different times experiments in three different times. I took 5 pictures in each group, then hide the group name and counted it by another observer for statistics. We also added this info in methods as suggested.

5. Ideally it would have been nice to see more experiments in addition to cell viability in hippocampal cells. The sensitivity to H2O2 and artemisinin was increased in these cells compared to SH-SY5Y cells, so do the authors expect that other results would be changed (e.g. AMPK pathway activation)?

Response: Thank the reviewer for the suggestion.  Although the sensitivity to H2O2 and artemisinin was changed, the protective effect is consistent. According to the reviewer, we also did some other experiments in Hippocampal neurons. The results were added in Fig.6 and supplementary data.

Reviewer 2 Report

The manuscript entitled " Artemisinin attenuated hydrogen peroxide (H2O2)-induced oxidative injury in SH-SY5Y and hippocampal neurons via the activation of AMPK pathway" aims to investigate the role of artemisinin as a potential therapeutic agent for neurodegenerative diseases. It is a very interesting research, well prepared, covering a lot of relevant information. Methodology and data analysis are very good, the writing style and the clarity of the exposition are accurate, conclusions are clear.

I have no specific scientific concern, I only suggest to update the reference list by carefully read and cite the following papers:

·       https://doi.org/10.1016/j.phrs.2018.03.012

·       DOI: 10.1016/j.fct.2018.08.001

Author Response

Response to reviewer 2 comments

Comments/Suggestions:

I have no specific scientific concern, I only suggest to update the reference list by carefully read and cite the following papers:  https://doi.org/10.1016/j.phrs.2018.03.01,   DOI: 10.1016/j.fct.2018.08.001

Response: Thank you very much for your support and the suggestion. We have read the paper and we have cited the two DOI in our manuscript. Please have a look on the manuscript

Reviewer 3 Report

Review report

In this manuscript submitted by Zhao et al., entitled “Artemisinin attenuated hydrogen peroxide (H2O2)- induced oxidative injury in SH-SY5Y and hippocampal neurons via the activation of AMPK pathway” reported that artemisinin protected SH-SY5Y and hippocampal neuronal cells from H2O2-induced cell death. The authors demonstrated that the artemisinin alleviated the H2O2-mediated effects via activating AMPK pathway.

However, the authors needs address the following comments.

Major comments

1.    How the arteminisinin executes protective effects via AMPK? Only by studying AMPK and p-AMPK, the authors cant concluded the AMPK involved, needs further validation for downstream or upstream signaling molecules.

2.     What is the rationale for choosing hippocampal neurons? Did the authors found the similar effects in other regional neurons?

3.    How the authors correlates this study with in vivo or clinical settings? If the authors use in vivo studies, it would be significant.

4.    Arteminisinin and its derivatives are metabolized into DHA, Did the DHA also will give the same effects? If so which will be more significant?  What is the bioavailabity of the Arteminisinin or DHA?

5.    In all the figures the scale bar is not mentioned clearly. Need to be uniform.

6.    In flowcytometry data, the apoptotic cells population only increased 0.7 to 2.3% and ART treatment brings into 1.4%. How many repeats were done? The data is significant?

7.    In Fig3a, the authors need to show the green channel alone also and why the mitochondrial membrane potential has been shown only SH-SY5Y. Not only mitochondrial membrane potential, all the experiments needs to be validated primary neurons.

8.    In some places the authors mentioned SH-SY5Y and in other palces SY5Y, follow the same.

Author Response

Response to reviewer 3 comments

Comments/Suggestions:

1.    How the arteminisinin executes protective effects via AMPK? Only by studying AMPK and p-AMPK, the authors cant concluded the AMPK involved, needs further validation for downstream or upstream signaling molecules.

Response: Thank you for your comments. Our mechanism is that oxidative stress leads to dysregulation of mitochondrial energy metabolism, further impairing mitochondria, thereby promoting the development and progression of neurodegenerative diseases (neural cell damage). Artemisinin can improve these damage by modulating the AMPK signaling pathway. AMPK was chosen because the AMPK signaling pathway is closely related to cell survival and mitochondrial function.  

We not only check the phosphorylation of AMPK, we also using AMPK inhibitor and SiRNA. Consistent with our AMPK data, AMPK inhibitor Compound C attenuated the protective effect of Artemisinin. Similar, depressing the expression of AMPKα by specific siRNA for AMPKα also inhibited the survival promoting effect of Artemisnin. These data put together indicated the important role of AMPK in the survival promoting effect of Artemisnin. Moreover, it is well known that nuclear factor erythroid 2-related factor 2 (Nrf2), the main switch for the expression of a majority of endogenous antioxidant enzymes also is involved. Our result showed that AMPK phosphorylated Nrf2 at the Ser550 residue. May be Nrf2 is the downsteam of AMPK. But we need to further verify. For the upstream and downstream molecules, we will do it carefully in future by checking its upstream kinases and using microarray according to reviewer suggestion.

2.     What is the rationale for choosing hippocampal neurons? Did the authors found the similar effects in other regional neurons?

Response: Hippocampal neuron is the most popular neuronal cultures using in evaluating the survival protection. The hippocampus is the main area of learning and memory, and the damage of neurons in the hippocampus can lead to a decline in learning and memory on Alzheimer's disease. That is why we choose hippocampal neurons. Similar results were obtained in cortex culture neurons.

3.    How the authors correlates this study with in vivo or clinical settings? If the authors use in vivo studies, it would be significant.

Response: Thank you for your comments. Yes, you are right. Oxidative stress is one of the main causes of neurodegenerative diseases such as Alzheimer's disease (AD). The pathogenesis of AD is not known but oxidative stress is one of the key hypotheses. The experiment in vivo is what we will do next. Alzheimer's disease is one of the major neurodegenerative diseases. We selected Alzheimer's disease transgenic model mice (3xTg mice), treated with artemisinin, taken mouse brain tissue and serum and detected oxidative damage indicators and disease-related indicators .Based on the oxidative stress hypothesis to check the therapeutic effect of Artemisinin on Alzheimer's disease.

4.    Arteminisinin and its derivatives are metabolized into DHA, Did the DHA also will give the same effects? If so which will be more significant?  What is the bioavailabity of the Arteminisinin or DHA?

Response: Thank you for the comments. In fact, we tried DHA also.  However, we found that DHA did not have significant protective effects under the same conditions as artemisinin. The protective mechanism of artemisinin may not through the pathway of metabolism to DHA. So artemisinin have more significant effect in our study.

5.    In all the figures the scale bar is not mentioned clearly. Need to be uniform.

Response: Thank you for your suggestions. We added scale bar in all the pictures as you suggested.

6. In flowcytometry data, the apoptotic cells population only increased 0.7 to 2.3% and ART treatment brings into 1.4%. How many repeats were done? The data is significant?

Response: We did this experiment three times. We had done this experiment with the help of ACCURI C6 plus cytometry. During this experiment we have set the 20000 events for each group, and the percentage of each quadrant is automatically counted during the collection process. After the experiment is over, the data is exported in the statistical part of the software. Statistical analysis was then performed using GraphPad Prism 5.0 statistical software.

7.  In Fig3a, the authors need to show the green channel alone also and why the mitochondrial membrane potential has been shown only SH-SY5Y. Not only mitochondrial membrane potential, all the experiments needs to be validated primary neurons.

Response: Thank you very much for the suggestion. As we know that the principle of JC-1 is the ratio of red/green fluorescence, so at the time of experiment we only saved merge pictures. And we also do the quantification on the basis of red/green ratio.  The experiments on hippocampal neurons were used to further validate the results in the SH-SY5Y cells. We repeat some of key experiments. As reviewer suggested that I need to do some other experiments in Hippocampal neurons. We already have done JC-1 and western blot in Hippocampal neurons and added result in Fig. 6 and supplementary data. We found that all the results were consistent.  

8. In some places the authors mentioned SH-SY5Y and in other palces SY5Y, follow the same.

Response: Thank the reviewer for the suggestion. We have unified the name of cells as SH-SY5Y as suggested.

Round 2

Reviewer 3 Report

The author responded all the comments raised by the reviewer in author’s response, did experiments in primary hippocampal neurons and included those results. Impressive results. However, some of the comments the authors need to include the data in at least supplement.

1.       The author stating in the first comment that our results showed that AMPK phosphorylated Nrf2 at the Ser550 residue. I could not find the results in the revised manuscript.

2.       In comment two, the author stating that similar results found in cortex neuron. I am not able to see the results.

3.       Authors stating that DHA didn’t show any significant effect. It would be interesting if the authors include those results. It is known that Artemisinin is less stable and it can be metabolized in to its derivatives including DHA. If the authors discuss all in discussion it would be interesting.

4.       The scale bar need to be mentioned with micrometer in the figure as well in legends.

Author Response

Response to reviewer comments

We are sincerely grateful to reviewer for pointing out the insufficiency in the paper and raising precious suggestions. We have revised on our manuscript according to your comments and a point-by-point response is provided. In order to facilitate your examination, all changes we made in the manuscript have been highlighted.

Comments/Suggestions:

1. The author stating in the first comment that our results showed that AMPK phosphorylated Nrf2 at the Ser550 residue. I could not find the results in the revised manuscript.

Response: Thanks a lot for your comments. According to the reviewer, the result was added in the supplementary data. We are sincerely grateful to reviewer for raising precious suggestions.

2. In comment two, the author stating that similar results found in cortex neuron. I am not able to see the results.

Response: Thanks a lot for your comments. According to the reviewer, the result was added in the supplementary data.

3.  Authors stating that DHA didn’t show any significant effect. It would be interesting if the authors include those results. It is known that Artemisinin is less stable and it can be metabolized in to its derivatives including DHA. If the authors discuss all in discussion it would be interesting.

Response: Thank you for your comments. The result added in the supplementary data. And we also discuss this in the revised manuscript, page9, line 221-225.

4. The scale bar need to be mentioned with micrometer in the figure as well in legends.

Response: Thank you for the comments.We mentioned scale bar in the figure legends as the reviewer suggested.